# Brain Natriuretic Peptide Protects Cardiomyocytes from Apoptosis and Stimulates Their Cell Cycle Re-Entry in Mouse Infarcted Hearts

**DOI:** 10.3390/cells12010007

**Published:** 2022-12-20

**Authors:** Anne-Charlotte Bon-Mathier, Tamara Déglise, Stéphanie Rignault-Clerc, Christelle Bielmann, Lucia Mazzolai, Nathalie Rosenblatt-Velin

**Affiliations:** Division of Angiology, Heart and Vessel Department, Centre Hospitalier Universitaire Vaudois and University of Lausanne, 1011 Lausanne, Switzerland

**Keywords:** heart regeneration, infarction, cardiomyocytes, brain natriuretic peptide, cell cycle, apoptosis

## Abstract

Brain Natriuretic Peptide (BNP) supplementation after infarction increases heart function and decreases heart remodeling. BNP receptors, NPR-A and NPR-B are expressed on adult cardiomyocytes (CMs). We investigated whether a part of the BNP cardioprotective effect in infarcted and unmanipulated hearts is due to modulation of the CM fate. For this purpose, infarcted adult male mice were intraperitoneally injected every two days during 2 weeks with BNP or saline. Mice were sacrificed 1 and 14 days after surgery. BNP or saline was also injected intraperitoneally every two days into neonatal pups (3 days after birth) for 10 days and in unmanipulated 8-week-old male mice for 2 weeks. At sacrifice, CMs were isolated, counted, measured, and characterized by qRT-PCR. The proportion of mononucleated CMs was determined. Immunostainings aimed to detect CM re-entry in the cell cycle were performed on the different hearts. Finally, the signaling pathway activated by BNP treatment was identified in in vitro BNP-treated adult CMs and in CMs isolated from BNP-treated hearts. An increased number of CMs was detected in the hypoxic area of infarcted hearts, and in unmanipulated neonatal and adult hearts after BNP treatment. Accordingly, Troponin T plasma concentration was significantly reduced 1 and 3 days after infarction in BNP-treated mice, demonstrating less CM death. Furthermore, higher number of small, dedifferentiated and mononucleated CMs were identified in adult BNP-treated hearts when compared to saline-treated hearts. BNP-treated CMs express higher levels of mRNAs coding for hif1 alpha and for the different cyclins than CMs isolated from saline-treated hearts. Higher percentages of CMs undergoing DNA synthesis, expressing Ki67, phospho histone3 and Aurora B were detected in all BNP-treated hearts, demonstrating that CMs re-enter into the cell cycle. BNP effect on adult CMs in vivo is mediated by NPR-A binding and activation of the ERK MAP kinase pathway. Interestingly, an increased number of CMs was also detected in adult infarcted hearts treated with LCZ696, an inhibitor of the natriuretic peptide degradation. Altogether, our results identified BNP and all therapies aimed to increase BNP’s bioavailability as new cardioprotective targets as BNP treatment leads to an increased number of CMs in neonatal, adult unmanipulated and infarcted hearts.

## 1. Introduction

Ischemia due to myocardial infarction (MI) induces the premature and sudden death of billions of cardiac cells (0.5–1 × 10^9^ cardiomyocytes (CMs) rapidly die when 20% of the myocardium is affected) [1,2]. This massive cell death immediately impacts heart function and contractility. At the cellular level, hypoxia-mediated death affects all cardiac cells, and followed by immune cell infiltration, development of CM compensatory hypertrophy and fibrosis. All these cellular adaptations will finally lead to heart failure.

In order to help patients to regain an active life as soon as possible after MI, it is essential to improve their cardiac function. As the number of CMs should negatively be correlated with the ejection fraction, the current treatments aim to increase the CM cell number by limiting their death and hypertrophy and by stimulating cardiac regeneration [3].

Cardiac regeneration is a physiological process, taking place during ageing in mammalian adult hearts in order to replace dysfunctional cells by new functional ones [4]. New CMs originate essentially in adult hearts from the proliferation of pre-existing CMs [3,5,6]. However, adult CMs have a very limited capacity to proliferate, which is not sufficient to restore normal heart function after heart injuries, such as MI. Unlike neonatal CMs, which have a high proliferation capacity (however lost 5–7 days after birth), DNA duplication in adult CMs leads mostly to binucleation instead of real cytokinesis [7,8]. Although hypoxia (in infarcted hearts) stimulates adult CM proliferation (about 3% of the CMs in the border area of infarcted hearts undergo proliferation), their level of proliferation does not reach that of newborn CMs [3,9,10]. In addition to the scarcity of adult proliferating CMs, identification of these cells is complicated by the fact that no marker really discriminates between CMs undergoing binucleation and cytokinesis and by the fact that proliferating CMs are preferentially localized in a specific area of adult hearts (i.e., in the subendocardial muscle of the left ventricle) [10]. Understanding the mechanisms responsible for the different levels of proliferation of newborn, adult “normoxic” and adult “hypoxic” CMs is however the key to improve cardiac regeneration.

These last several years, we studied the cellular mechanisms responsible for the cardioprotective effects of the Brain Natriuretic Peptide (BNP) in infarcted hearts [11,12,13]. BNP, synthetized by ventricular CMs, belongs, with the atrial natriuretic peptide (ANP) synthetized by atrial CMs, and the C-type natriuretic peptide (CNP) synthetized by endothelial cells, to the natriuretic peptide (NP) family. NPs bind with different affinity to three receptors, NPR-A, NPR-B and NPR-C. BNP binding to NPR-A or NPR-B increases the cGMP intracellular concentration and activates the protein Kinase G [14]. CMs, cardiac fibroblasts and endothelial cells in neonatal and adult hearts express these three receptors and are thus able to respond directly to NP stimulations [15].

Interestingly, NP heart concentrations evolve with ageing and cardiac pathologies. ANP and BNP are abundantly expressed during embryogenesis and their levels decrease after birth [15]. Their expressions however transiently increase 3 and 5 days after birth in neonatal hearts at time where postnatal CMs re-enter into the cell cycle and proliferate [16]. During development, NPs modulate essentially CM proliferation [17]. In adult “healthy or unmanipulated” hearts, despite reduced levels, NPs contribute to heart homeostasis by controlling fibrosis, CM hypertrophy and contractility [18]. The role of NPs on CM contractility is not fully elucidated but is clearly dependent on cGMP compartmentation [18,19]. Indeed, an increased cGMP level induced by BNP or CNP stimulations does not have the same effect on CM contractility [19]. cGMP compartmentation is a key mechanism, which depends on the localization of the NP receptors and phosphodiesterases, able to degrade the cGMP into its inactive form. NPR-A receptors are found in transverse T-tubules, associated with phosphodiesterase 2, which leads to restricted cGMP diffusion [20]. NPR-B receptors are uniformly localized on the CM membrane and produces diffusible cGMP signals [18]. Thus, CNP binding to NPR-B affects CM contractility, whereas BNP, which acts via NPR-A, has no or little effect on CM contractility [19].

In “stressed” hearts (i.e., in ischemic hearts or in hearts submitted to mechanical stretch induced by volume or pressure overload), CMs secrete high levels of BNP. However, because of O-glycosylation, secreted BNP is to a large extent biologically inactive [21,22]. Thus, BNP supplementation has been shown in several animal models of ischemic hearts to be beneficial by limiting CM hypertrophy and fibrosis [11,15,23,24,25,26,27].

The outcome of BNP therapy in patients suffering from cardiovascular diseases is unclear. The first clinical trials which were performed using Nesiritide (recombinant human BNP), highlighted positive hemodynamic and clinical effects in patients with acute heart failure, but also severe adverse effects, such as hypotension, renal failure and a higher mortality rate in the group of BNP-treated patients [28]. Thereafter, the use of BNP dramatically decreased in clinics [29]. However, other clinical studies demonstrating that low doses of Nesiritide administered to patients with acute heart failure improves cardiac function without inducing hypotension, nephrotoxicity or increasing the rate of death or rehospitalization, allows for the debate about the usefulness of BNP therapy for patients with heart failure to reopen [28,30,31,32,33]. In particular, subcutaneous injection of BNP in patients with chronic systolic heart failure was shown to have cardiac beneficial effects without renal toxicity and hypotension [33]. Accordingly, published meta-analysis suggests a protective role of ANP/BNP infusion in patients with acute myocardial infarction [34]. Recently, a randomized clinical trial showed that 3-day-long infusion of BNP in patients with an anterior myocardial infarction was not sufficient to positively modify the parameters of the left ventricular remodeling. However, patients treated with BNP who had a baseline left ventricular ejection fraction of less than 40% showed a reduced left ventricular infarction size compared to the placebo [35].

Another therapeutical option able to increase BNP bio-availability is currently being studied. Indeed, great interest is focused on the LCZ696 treatment (Entresto), which associates both an angiotensin receptor blocker (valsartan) and an inhibitor of neprilysin (NEP, sacubitril). NEP is an endopeptidase able to degrade several factors such as the natriuretic peptides (ANP, BNP and CNP) but also angiotensin II, bradykinin, endothelin-1 [36]. Treatments of mice, rats, rabbits and humans with an NEP inhibitor increase the blood level of BNP and we and others measured increased levels of ANP, BNP and cGMP in the plasma of patients, rats and mice treated with LCZ696 [13].

Thus, the effects of BNP treatment in humans are less conclusive than in animal models of heart injuries. This results mainly from the lack of data about the real role of BNP in infarcted and healthy hearts. New data will help to adapt BNP concentrations, way of administration, timing in order to improve its effect in patients suffering from cardiovascular diseases.

In previous work, we highlighted the cardioprotective effect of BNP intraperitoneal (ip) injections in infarcted mice [11,13]. Contractility increased twofold and heart remodeling (the percentage of changes of the left ventricle volume) decreased (−79%, *p* = 0.04) 4 weeks after MI in BNP-treated mice when compared to saline-injected mice [11]. At the cellular level, BNP injections decreased the infarct size by 20% and reduced fibrosis [13]. At least a part of the BNP positive effect in infarcted mice is due to increased cardiac angiogenesis and vasculogenesis [13]. Indeed, BNP stimulates the proliferation of both endogenous endothelial cells and endothelial precursor cells, which accelerates the heart revascularization after MI [13].

Because CMs express BNP receptors, we questioned in this work whether BNP is also able to modulate CM’s fate in adult infarcted hearts.

This work aimed to determine whether BNP treatment in infarcted hearts modulates CM apoptosis, hypertrophy, binucleation and proliferation. Furthermore, BNP effect on CM proliferation was compared between neonatal, adult “normoxic” and “hypoxic” CMs, which display high, low and medium proliferation capacity, respectively.

## 2. Results

### 2.1. The PKG Signaling Pathway Activated in Cardiomyocytes after BNP Intraperitoneal Injections

CMs isolated from neonatal and adult hearts express BNP receptors (Appendix A), suggesting that they could respond directly to BNP stimulations. NPR-A and NPR-B are equally expressed in neonatal CMs, whereas adult CMs express mainly NPR-A.

As previously reported, BNP binding to NPR-A or NPR-B increases the cGMP level, PKG activation and phospholamban (PLB) phosphorylation [37]. Phosphorylated PLB related to the total amount of PLB was thus used as a marker of BNP binding to NPR-A or NPR-B. Following BNP injections, pPLB/PLB ratio increased in CMs isolated from the border zone (BZ) of infarcted hearts (×2.7), from unmanipulated neonatal (×3.5) and adult (×1.5) hearts but decreased (−32%) in the CMs isolated from the RZ of infarcted hearts (Appendix A), demonstrating that BNP ip injections act on CMs.

### 2.2. More Cardiomyocytes after BNP Injections

BNP injections did not change the mouse mortality at 14 days. 83.9% and 84.2% of the saline or BNP-injected infarcted mice, respectively, survived. The number of CMs in the hearts was evaluated after BNP treatment (Figure 1). CMs were isolated from the hearts after enzymatic digestion using the Langendorff-Free method and counted [38]. CMs were identified by their size and by their expression of Troponin I. Both rod-shaped and small round CMs were counted.

Following BNP treatment, the total number of CMs per heart was increased by 81% in the infarct area (ZI) of infarcted hearts, and by 18 and 19% in unmanipulated neonatal and adult hearts, respectively, when compared to the numbers in saline-injected mice (Figure 1). No increase in the CM cell number was detected in the BZ or RZ of infarcted hearts. To confirm this result in adult unmanipulated hearts, Myh6 MerCreMer mice were injected with tamoxifen (to induce the expression of the GFP protein in the CMs) and with BNP 2 weeks later. Two weeks after the first BNP injection, CMs were identified as Troponin I^+^ GFP^+^ cells by flow cytometry analysis and counted. The number of Troponin I^+^ GFP^+^ cells increased by 28% in BNP-treated hearts when compared to the number in saline-treated hearts (Appendix A).

The size of the heart depends on the age and the size of the different pieces of the infarcted hearts varies between the dissections. To normalize these parameters, the CM numbers were also related to the weight of the hearts or pieces of infarcted hearts (Appendix A). The number of CMs per mg tissue increased 2-fold in the ZI of infarcted hearts, and increased by 44 and 26% in neonatal and adult hearts, respectively, after BNP treatment. No change of the CM number per mg tissue in the BZ and RZ of infarcted hearts.

Finally, we also related the number of CMs in BNP-treated hearts to the number of CMs in saline-injected mice. The BNP and saline-treated mice were littermates, injected during the same experiment and CMs were isolated the same day with the same enzymatic cocktail. The number of CMs increased 2.2-fold in the ZI of BNP-treated infarcted hearts when compared to the number of CMs isolated from the same area of saline-treated hearts. Similar numbers of CMs were isolated from the BZ and RZ of BNP- and saline-treated hearts (Appendix A). In unmanipulated neonatal and adult hearts, BNP treatment increased the number of CMs by 23 and 28.5%, respectively, when compared to the numbers in saline-treated hearts.

Interestingly, BNP treatment did not change the CM cell number in NPR-A deficient hearts, demonstrating that the BNP effect on the CM cell number is dependent on NPR-A expression (Figure 1C and Appendix A).

### 2.3. BNP-Treated Cardiomyocytes Are Smaller, More Mononucleated and Express Higher mRNA Level Coding for Hif1α

CMs isolated from BNP- and saline-treated infarcted or unmanipulated neonatal and adult hearts, were characterized for their mRNA profile, nucleation status and size (Figure 2).

Adult CMs isolated either from the border area of infarcted hearts (Figure 2A) or from unmanipulated hearts (Figure 2F) expressed, after BNP treatment, increased levels of mRNAs coding for the fetal isoform of myosin heavy chain (*myh7*) (×2.7 and ×1.9, respectively), for the alpha skeletal actin (*acta1*) (×2 and ×1.4, respectively) and for the hypoxic inducible factor-1α (*hif1*α) (×1.4 and ×1.8, respectively). No difference in the mRNA expression of these genes was highlighted in CMs isolated from the RZ of BNP-treated hearts (data not shown). mRNA level coding for *hif1*α was also increased in CMs isolated from neonatal BNP-treated hearts (×1.5) (Figure 2D).

The percentages of mononucleated CMs increased in all the BNP-treated hearts (Figure 2B,E,G). The frequency of small CMs was significantly higher in the CMs isolated from all the area of the BNP-treated infarcted hearts than in CMs isolated from saline-injected infarcted hearts (Figure 2C). We already showed in previous work that adult CMs isolated from unmanipulated BNP-treated hearts also displayed a reduced cross-sectional area (−6%, *p* = 0.008) when compared to CMs isolated from saline-injected hearts [11].

Altogether these results demonstrate that BNP treatment in adult hearts leads to an increased number of small CMs and to higher percentages of mononucleated CMs. Immunostainings showed that the small CMs were mostly mononucleated (Figure 3A,B,E). Furthermore, BNP-treated CMs express higher levels of *myh7*, *acta1* and *hif1*α than CMs isolated from saline-treated hearts.

### 2.4. BNP Treatment Protects All Cardiac Cells from Cell Death in Infarcted Area

To identify the mechanisms leading to the increased number of CMs in the infarcted area of BNP-treated infarcted hearts, we first evaluated whether BNP is able to protect the CMs from cell death. Troponin T concentrations were measured 1 and 3 days after MI in the plasma of infarcted mice treated or not with BNP (Figure 4A). Circulating Troponin T is a marker of myocardial cell damage and predicts adverse outcomes in patients with heart failure [39]. Troponin T plasma concentrations were significantly reduced after BNP treatment (×2.6 and 2.1 decreased 1 and 3 days after MI, respectively). Interestingly, the caspase 3 protein level was decreased (−42%) in the ZI of BNP-treated hearts one day after MI (Figure 4B), demonstrating that less cardiac cells undergo apoptosis in the ZI of BNP-treated hearts compared to saline-treated hearts. Immunostainings aimed to identify apoptotic CMs 1 day after MI showed however only a slight difference in the number of caspase 3^+^ CMs (−13%) in the infarcted area of BNP-treated hearts compared to saline-treated ones (Figure 4C,D).

### 2.5. BNP Stimulates Cardiomyocyte Cell Cycle Re-Entry

Adult CMs undergoing a real cytokinesis are rare in unmanipulated hearts. However, in the BZ of infarcted hearts or in hearts injected with different factors, such as FGF10 or Neuregulin 1, CMs can re-enter the cell cycle and proliferate [6,40,41,42]. CMs re-entering the cell cycle to undergo binucleation or cytokinesis, were identified by several authors as being small mononucleated CMs expressing HIF1α [43,44]. We thus evaluated whether BNP is able to stimulate CMs to re-enter the cell cycle in adult infarcted hearts as well as in neonatal and adult unmanipulated hearts. We assessed by immunostainings the expression by CMs of different markers of the cell cycle, such as Ki67, expressed in all phases of the cell cycle, phosphorylated histone H3 (pH3), expressed during the G2/M phase, and Aurora B (Aurkb) expressed during the M, G2 and cytokinesis phases [45]. Furthermore, we discriminated between Aurkb^+^ CMs in prophase (localization of Aurora B in the nuclei) and in binucleation or cytokinesis (localization in the cytoplasm). The detection of BrdU incorporation in CMs identifies cells undergoing DNA synthesis (S phase). The mRNA expressions of the different cyclins involved in the cell cycle (cyclin D1 to Cyclin B2), with a focus on A2 and B2 cyclins expressed during cytokinesis, were measured in isolated CMs.

Higher percentages of CMs incorporating BrdU (+70%), or expressing Ki67 (+50%), pH3 (+40%) or Aurkb (×2.8) were detected in the infarct and border area of BNP-treated hearts 10 days after MI when compared to the percentages obtained in saline-infarcted hearts (Figure 3A–C,E,F). Accordingly, in the ZI+BZ of infarcted Myh6 MerCreMer mice injected with tamoxifen, BNP treatment increased the percentages of BrdU^+^ GFP^+^ α actinin^+^ cells among all GFP^+^ α actinin^+^ cells (+20%) (Figure 3D). CMs incorporating BrdU or expressing pH3 or Aurkb were mainly small dedifferentiated (i.e., α actinin protein was disorganized) mononucleated CMs (Figure 3A(c), 3B(b,c) and 3E(a,b)). Aurora B was however also detected in binucleated CMs (Figure 3E(c,d)). In these latter cells, Aurora B localization was cytoplasmic and not nuclear, suggesting that these cells underwent binucleation or cytokinesis. Among Aurkb^+^ CMs, the number of CMs undergoing cytokinesis or binucleation slightly increased after BNP treatment (Figure 3F).

All CMs isolated from the BZ of the BNP-treated hearts, except those from heart 1, showed an increased level of mRNAs coding for at least one cyclin (Figure 3G). In some hearts, we detected increased levels of mRNAs encoding different cyclins, demonstrating that CMs were not synchronized within one heart and that different CM subsets may exist at different steps of the cell cycle (mouse 3, 4, 7, 8). The levels of mRNAs coding for cyclin A2 and B2 were increased in the CMs isolated from 4 out of 8 hearts.

In neonatal BNP-treated hearts, the numbers of CMs incorporating BrdU, or expressing Ki67, or pH3 or Aurkb were increased after BNP treatment (+23, 15, 41 and 47.5%, respectively) (Figure 5A–E). The number of Aurkb^+^ CMs undergoing cytokinesis or binucleation in BNP-treated hearts increased 1.4-fold when compared to saline-treated hearts (Figure 5E). Furthermore, we identified CMs undergoing cytokinesis in BNP-treated hearts by measuring the distance between both nuclei of Ki67^+^ CMs expressing Aurkb (Figure 5C) [46]. Isolated CMs from all BNP-treated hearts displayed increased levels of mRNAs coding for cyclins E1, A2 or B2 (Figure 5F).

In unmanipulated adult hearts, higher percentages of CMs incorporating BrdU (+67%) or expressing pH3 (+94%) were detected after BNP treatment (Appendix A). As in adult CMs isolated from the BZ of the infarcted hearts, the expressions of the mRNAs coding for the different cyclins in the isolated CMs were very heterogeneous from one BNP-treated heart to another (Appendix A). However, the levels of the mRNA coding the cyclin A2 or B2 increased in 5 out of 7 CM isolations.

### 2.6. BNP Direct Effect on Cardiomyocytes

The effect of BNP treatment on CMs results from direct and/or indirect interactions. The improvement in vascularization in infarcted hearts after BNP treatment could for example increase the survival of the CMs [13]. In vitro studies on CM cell cultures were performed to study whether BNP can act directly on CMs. CMs were isolated from neonatal and adult C56BL/6 hearts and treated in vitro with BNP (Appendix A).

Neonatal CMs were cultured with three different BNP concentrations (10, 100 and 1000 nM) (Appendix A). The number of CMs was increased in cell cultures treated with 10 nM BNP (+18%) (Appendix A). A higher number of small and mononucleated CMs was detected in BNP-treated cell cultures (Appendix A). BNP treatment increased the percentages of CMs expressing Ki67 (+23%) or Aurkb (+50%) (Appendix A). Neonatal CMs stimulated in vitro with BNP expressed high levels of mRNAs coding for cyclin D1 (+50%), A2 (+40%) and B2 (+30%) (Appendix A). Additionally, CM cell division was also followed using time-lapse microscopy on neonatal CMs isolated from Myh6MerCreMer hearts and treated with 10 nM BNP (see movie, Appendix A).

Adult CMs were isolated from 6–8-week-old C57BL/6 hearts and cultivated for 7 days with or without BNP (10 nM) (Appendix A). After 7 days of culture, mRNA levels coding for anf (+70%), dab2 (+52%), runx1 (+62%) and cyclin E1 (+56%) were increased in BNP-treated CMs compared to untreated ones. More interestingly, we detected only in BNP-treated cell cultures Aurkb^+^ Troponin I^+^ CMs.

### 2.7. BNP Acts on Adult CMs via ERK MAP Kinase Activation

Finally, we identified the signaling pathway activated by BNP in adult CMs. Adult CMs were isolated from C57BL/6 hearts and stimulated or not with BNP for 2–3 h. Increased pPLB/PLB (×2.1), pAkt/Akt (×2.5) and pERK/ERK (×1.9) ratios were measured in BNP-treated cells compared to untreated ones (Figure 6A). pp38/p38 ratio did not change. We also isolated CMs from BNP and saline-treated hearts (i.e., BNP or saline was injected 2–3 h before CM isolation). As already presented in Appendix A, the pPLB/PLB ratio was increased (×1.5) in CMs isolated from BNP-treated mice. BNP treatment in vivo also increased the pERK/ERK (×1.6) ratio (Figure 6B).

As BNP stimulates the phosphorylation of ERK in adult isolated CMs in vitro and in adult CMs in vivo, we thus tested whether ERK phosphorylation via BNP, stimulates CMs to re-enter into the cell cycle. PD0325901 (an ERK inhibitor) was ip injected in unmanipulated adult mice 1 h before BNP injection for 2 weeks. In the presence of ERK inhibitor, BNP treatment did not increase the number of CMs in adult unmanipulated hearts (Figure 6C).

### 2.8. Increased Number of CMs in Infarcted Hearts after LCZ696 Treatment

BNP injections in human patients suffering from cardiovascular diseases are not as beneficial as in animal models. Increasing BNP concentration by inhibiting its degradation is another way to increase BNP bioavailability in patients. This can be performed by treating patients with LCZ696 (Entresto, Novartis). We treated infarcted mice with LCZ696 for 10 days [13] (Figure 7A). Ten days after infarction, we determined in LCZ696 treated mice an improvement of the cardiac function (fractional shortening: ×2; ejection fraction: ×1.9, with 60 mg/kg/day LCZ696) and a 2-fold decrease in the heart remodeling [13]. mRNA levels coding for *troponin I* (+40%) and *myh6* (+60%) were increased in the ZI+BZ area of LCZ696-treated hearts 10 days after infarction, suggesting higher number of cardiomyocytes in these areas when compared to untreated infarcted hearts (Figure 7B). Accordingly, the number of CMs increased in all areas of the infarcted hearts treated with LCZ696 (+34, +17 and +11%, in ZI, BZ and RZ, respectively) (Figure 7C,D). No difference between both concentrations of LCZ696 was detected. Higher percentages of CMs in the ZI+BZ of LCZ696 treated hearts incorporated BrdU (+61%) and expressed Aurkb (+85%) when compared to the percentages of untreated infarcted hearts (Figure 7E). Interestingly, BrdU^+^ CMs were small mononucleated and dedifferentiated CMs (Figure 7C, white arrows). Our results suggest that LCZ696 stimulates also CM proliferation. Whether this is due to increased BNP level remains to be proved.

## 3. Discussion

Although numerous works highlighted the BNP cardioprotective role in rodent infarcted hearts, the precise cellular mechanisms by which BNP increases heart function and decreases heart remodeling, are not precisely identified yet [11,19,23,24,25,26,33]. In this report, we investigated whether a part of the cardioprotective effect of BNP is due to the modulations of the CM’s fate. Therefore, we studied the consequences of BNP treatment on CM survival and renewal in three different conditions: in adult infarcted hearts and in unmanipulated neonatal and adult hearts. In the adult infarcted hearts, CMs are submitted to hypoxia-leading cell death in the infarct and border zone and to increased workload leading to CM hypertrophy in the border and remote area. Neonatal CMs are small round cells, mostly mononucleated, exhibiting a glucose based-metabolism. Adult CMs are elongated cells, with increased intracellular myofilaments, sarcoplasmic reticulum and mitochondria. They are mainly bi-or polynucleated and dependent on fatty acid metabolism. Both neonatal and adult CMs express BNP’s receptors, which would not prevent them from responding differently to BNP stimulation depending on their environment.

However, this was not the case. Surprisingly, BNP treatment resulted in higher CM cell number than saline-treatment, whatever the heart environment. Therefore, it remained to identify the mechanism(s) by which BNP treatment increases the number of CMs in the neonatal and adult unmanipulated and infarcted hearts. Two mechanisms have been evaluated: (1) BNP-mediated CM protection by directly inhibiting CM apoptosis or by inducing DNA synthesis and/or (2) improvement of CM renewal by the stimulation of their proliferation.

The originality of our work is that we isolated, counted and compared the CMs from BNP and saline-treated hearts in pathological and physiological conditions. qRT-PCR, flow cytometry analysis and Western blot analysis were directly performed on isolated CMs and not on heart tissues. This methodology could also be a limitation as CM isolations were performed using enzymatic digestion, which could alter the CM capacity to survive after isolation [8]. We thus wondered if the higher number of CMs isolated from BNP-treated hearts is the consequence of the BNP treatment in vivo, which could render the CMs more resistant to enzymatic digestion during isolation. This is, however, unlikely.

Indeed, if BNP protects CMs from enzymatic digestion, we should have an increase in the CM number in all areas of the infarcted hearts, which is not the case. However, to test this hypothesis, we determined by flow cytometry analysis the percentages of apoptotic CMs (i.e., DAPI^+^ Troponin I^+^ cells) in CMs isolated from adult hearts injected during 2 weeks with saline or BNP. The percentage of DAPI^+^ CMs was the same within CMs isolated from adult BNP- (4 ± 1%) or saline (3.5 ± 0.5%)-treated hearts, demonstrating that BNP treatment does not protect CMs from death induced by enzymatic digestion. Furthermore, these results obtained after enzymatic digestion of the hearts, corroborate our previously published results where we detected using immunostainings (with anti-laminin and anti-Nkx2.5 antibodies) an increase in the Nkx2.5^+^ myocytes in neonatal (+22.5%) and adult (+68%) BNP-treated hearts [11].

Several studies, performed essentially in vitro, evaluated the role of BNP on CM apoptosis with contrasted results [26,47,48,49,50]. Neonatal rat CMs in culture are protected against apoptosis by BNP treatment after ischemia/reperfusion [47,48,50], whereas BNP-treated human CMs increased their lncRNA LSINCT5 expression, which leads to increased apoptosis [49]. Furthermore, Gorbe et al. showed that this BNP protective effect is mediated via PKG activation in neonatal rat cardiac myocytes [50]. In vivo, in rabbit hearts, BNP treatment protects all cardiac cells from apoptosis after ischemia/reperfusion [26].

Concerning the infarcted hearts, CM cell death at the early phase takes place during the first 6–24 h after ischemia [51]. The two main mechanisms responsible for cell death after MI remain apoptosis and necrosis [52]. In our study, we measured a reduced level of Troponin T in the plasma of BNP-treated mice 1 and 3 days after MI, which suggests that less CMs died in these mice. Accordingly, the results of Western blot analysis showed that the caspase 3 protein level decreased in all cardiac cells of the infarction area in BNP-treated hearts. However, we did not detect using immunostainings a significant decrease in the number of apoptotic CMs in BNP-treated hearts 24 h after MI. This result is probably due to the limitation of the technic (not enough dying cells to detect a difference) or to the fact that BNP could also protect CMs undergoing other cell death mechanisms, such as necrosis, necroptosis and/or autophagy.

BNP could also protect CMs in infarcted hearts from death occurring after this early phase. Indeed, a low number of CMs continues to die in infarcted hearts until heart failure [53]. We thus cannot exclude that BNP treatment protects also the CMs during this late phase of cell death, which leads at long term to an increase in the number of CMs. This could also be the case in adult unmanipulated hearts where BNP could protect CMs from “physiological” death occurring during normal life [4,5].

BNP can also “indirectly” protect CMs from cell death by making them more resistant to the stress. It was shown by Jiang et al. that polyploid/multinucleated CMs are more resistant to stress and have the ability to maintain heart contractility after injury compared to mononucleated diploid CMs [54]. Furthermore, hypoxia on mice triggers an increased number of polyploid and multinucleated CMs, which display a better adaptation to stress by decreasing apoptosis and ROS production compared to mononucleated/diploid CMs [54]. In this case, the increased number of CMs expressing Ki67, pH3 and Aurora B detected in our work, could be a protective mechanism, leading to DNA synthesis and binucleation in our CMs. However, in BNP-treated hearts, the percentages of mononucleated CMs increased, suggesting rather a stimulation of the CM cytokinesis than of the binucleation process.

Neonatal CMs have a high capacity to proliferate, which allow them to participate actively in heart regeneration and repair [7]. In our work, BNP undoubtedly stimulates the proliferation of the neonatal CMs in vivo and in vitro. Seven to ten days after birth, heart regeneration is drastically decreased and a few damaged CMs are replaced by new functional ones. Fibrosis develops and heart contractility decreases. This is essentially due to the very limited capacity of the majority of the adult CMs to proliferate [1,3]. However, in adult hearts, a subset of small mononucleated CMs survives and are able to proliferate, participating thus to heart homeostasis [5,6,42,55,56]. In hypoxic conditions, these proliferative CMs express *hif1 alpha* and participate to heart regeneration [43,44]. Factors, such as Neuregulin 1, or FGF-10 stimulate their proliferation that results in improvement of heart repair [40,41,42].

Our results suggest that BNP also stimulates the proliferation of these small mononucleated CMs. Indeed, an increased percentage of small CMs was detected in BNP-treated adult hearts. Decreased CM size was often associated with increased proliferation in other works [6,40,42,55]. In our case, the decreased size of the CMs did not result from the anti-hypertrophic effect of the BNP, as the mRNA level coding for *anf* did not change between saline- and BNP-treated CMs. Thus, a higher number of small CMs may result from the proliferation of CMs, which dedifferentiate before dividing [41,57,58,59]. Immunostainings using an antibody against alpha actinin showed clearly that adult BrdU^+^, Ki67^+^ or pH3^+^ CMs were small CMs with disorganized alpha actinin structure. Furthermore, CMs isolated from BNP-treated infarcted hearts expressed higher levels of mRNAs coding for *myh7* and *acta1*, both genes re-expressed during dedifferentiation process, than CMs isolated from saline-injected hearts. In vitro, adult BNP-treated CMs re-expressed *runx1* and *dab2*, known markers of the dedifferentiation process, and Aurkb^+^ cardiomyocytes showed disorganized Troponin I structure.

The results obtained with the LCZ696 treatment are also very interesting. LCZ696 treatment started 1 day after MI, after the acute phase of cell death and led however to increased numbers of CMs in the ZI+BZ and even in the RZ of infarcted hearts. This demonstrates that the BNP effect in infarcted hearts was not only due to the protection against apoptosis in the first 24 h but resulted clearly also either from late “CM protection” or from increased CM renewal. As in BNP-treated hearts, we detected an increased number of small dedifferentiated BrdU^+^ CMs in LCZ696-treated hearts, suggesting that LCZ696 stimulated the CMs to dedifferentiate and to proliferate. LCZ696 treatment in patients and in animal models was shown to increase the level of BNP, but not only that. Increased levels of ANP were also detected and could contribute to the stimulation of the CM proliferation [13].

Finally, we investigated by which signaling pathway BNP modulates CM cell number. Increased pPLB/PLB ratio was measured in CMs isolated from the BZ of BNP-treated infarcted hearts and from BNP-treated neonatal and adult hearts. In vitro BNP-treated adult CMs showed also increased pPLB/PLB ratio, demonstrating that BNP can bind directly to NPR-A or NPR-B on CMs. Thanks to NPR-A deficient mice, we demonstrated that BNP stimulates via NPR-A the proliferation of adult CMs.

We then showed that BNP treatment leads to activation of the MAPK/ERK pathway in adult CMs stimulated in vitro, or in vivo. This pathway plays a central role in cardiac physiology by modulating cell proliferation, cell growth and hypertrophy [60,61,62]. After phosphorylation, ERK translocates into the nucleus and upregulates the transcription of cyclin D1 and D2 and thus stimulates the transition from G1 to S phase [63,64]. Activation of the MAP/ERK pathway was shown to modulate CM dedifferentiation and proliferation in several heart conditions and especially in adult mouse infarcted hearts stimulated by Neuregulin/Erbb2 activation, a potent activator of the adult CM dedifferentiation and proliferation [41,61,62,65,66,67]. Interestingly, Strash et al. recently showed that ERK activation via the Neuregulin Erbb2 receptor stimulates the dedifferentiation and proliferation of human induced pluripotent stem cell-derived CMs and neonatal rat ventricular myocytes in vitro [68].

This work identifies for the first time the mechanisms by which BNP contributes to adult heart homeostasis in physiological conditions and to the protection of the adult hearts after MI. In both conditions, BNP treatment leads to increased CM cell number by protecting the CMs from cell death and stimulating the proliferation of small mononucleated CMs. These results open the door to new treatments aimed at improving CM protection and replacement.

## 4. Material and Methods

### 4.1. Mice Strains

All animal procedures were approved in accordance with the recommendations of the U.S. National Institutes of Health Guide for the Care and Use of Laboratory Animals (National Institutes of Health publication 86–23, 2011). All experiments were approved by the Swiss animal welfare authorities (authorizations VD2765, VD28651 and VD3211) and conform to the guidelines from Directive 2010/63/EU of the European Parliament on the protection of animals used for research. C57BL/6 mice [6] (Wild Type mice, WT) were purchased from Janvier (Le Genest-Saint-Isle, France). Myh6 MerCreMer mice (JAK-5657) and Tomato-EGFP mice (JAK-7576) were purchased from the Jackson Laboratory (Bar Harbor, Main, US). Heterozygous Myh6 MerCreMer/Tomato-EGFP adult mice were bred in our animal facility. The NPR-A (−/−) mice were kindly provided by Dr Feng Li and Prof Nobuyo Maeda (Chapel Hill, NC, USA) [69]. Only male mice were used.

### 4.2. In Vivo Procedures

#### 4.2.1. Tamoxifen Injections in Adult Myh6 MerCreMer Mice

Before surgery and/or BNP treatments, adult Myh6 MerCreMer mice (6-week-old) were injected with Tamoxifen (Sigma, Merck, Darmstadt, Germany, T5648) at 40 mg/kg to activate the Cre recombinase. Tamoxifen was dissolved in ethanol to a concentration of 100 mg/mL and emulsified in peanut oil to a final concentration of 10 mg/mL. A total of 1 mg Tamoxifen/25 g body weight was injected intraperitoneally (i.p) to adult mice one time. Two weeks after tamoxifen injection, 90–95% of pre-existing CMs expressed the GFP protein [11].

#### 4.2.2. Surgery Leading to Myocardial Infarction

To generate a permanent infarction, a ligation of the left anterior descending coronary artery (LAD) was performed in 6–8-week-old C57BL/6- or 8-week-old Myh6 MerCreMer mice, already injected with Tamoxifen 2 weeks before. Mice were anesthetized with ketamine/xylazine and acepromazine (65 mg/kg, 15 mg/kg, 2 mg/kg, respectively). Mice were intubated and ventilated during all the entire period of surgery. At the third intercostal space, the chest cavity was opened and at the left upper sterna border, the LAD coronary artery was ligated with a 7–0 nylon suture at about 1–2 mm from the atria [13]. After surgery, Temgesic (Buprenorphine, 0.1 mg/kg) was injected subcutaneously as soon as the mice woken up and every 8–12 h for 2 days. Adult mice were sacrificed by lethal injection of pentobarbital (150 mg/kg intraperitoneally) 2 weeks after the surgery. The breathing arrest was controlled and cervical dislocation was performed.

To assess for the reproducibility of LAD occlusion, we measured the infarct size by echocardiography analysis (% of the MI length long axis). Two weeks after surgery, the infarct area represented 37 ± 2.5% of the LV in saline injected mice (n = 10) and 31.5 ± 4% in BNP-treated mice (n = 7). BNP reduced the infarct size by 15% as already demonstrated in our previous work [11,13].

#### 4.2.3. In Vivo BNP Injection

BNP was injected into infarcted adults, and unmanipulated neonatal and adult mice. Directly after surgery, NaCl (saline) or BNP (1.3 µg/20 g in 20 µL) was injected into the left ventricle cavity. At 24 h after MI, mice were intraperitoneal (ip) injected with BNP (2 µg/20 g mice) or saline every 2 days [11]. C57BL/6 mice were sacrificed 1- and 14 days post-MI and Myh6 MerCreMer mice were sacrificed 14 days post-MI (Figure 1A). Neonatal C57BL/6 mice (3 days after birth) were injected ip with NaCl or BNP every two days (1 µg/2 g; Bachem; synthetic mouse BNP (1–45) peptide (catalog number H-7558)). Mice were sacrificed 11 days after birth (Figure 1B). Eight week old C57 BL/6 mice or Myh6 MerCreMer unmanipulated mice were injected ip with BNP (2 µg/20 g mice) or NaCl every two days (Figure 1C) [11]. Two weeks after the first injection, mice were sacrificed. To assess for the implication of the ERK signaling pathway in the BNP effect on CMs, PD0325901 (60 µg/20 g mice in 80 µL) was administered by gavage in adult unmanipulated mice 1 h before BNP injection (2 µg/20 g mice).

For all experiments with adult mice, BrdU (1 mg/mL) was added into the drinking water during the entire period of the experiments. BrdU (40 µL/mice, 10 mg/mL) was ip injected in newborn mice on day 5 and 9 after birth.

The concentrations of BNP used in this study were already shown to stimulate the mobilization, proliferation and differentiation of the endothelial precursor cells [70,71]. In our first work, we verified that these concentrations have no impact on blood pressure, heart rate and cardiac output [11]. In contrast, these BNP concentrations decrease the heart remodeling, the left ventricular volume and left ventricular end diastolic diameter and increase by 2-fold the contractility, 4 weeks after MI [11]. Furthermore, at the cellular level, these BNP concentrations increase the number of cardiac precursor cells (i.e., Sca-1^+^ cells), the number of endothelial precursor cells (WT-1^+^ cells) and stimulate the proliferation of the endothelial cells [11,12,13]. All these mechanisms lead to increased re-vascularization of the ischemic area in BNP-treated infarcted mice [13].

#### 4.2.4. BNP Kinetic In Vivo

The half-life of BNP is about 18 min in the circulation of healthy persons and 12 min in the plasma of patients with congestive heart failure [72,73]. In rats with myocardial damage, the BNP half-life is about 1.2 min, which is very short [74]. In previous work, we followed the BNP kinetic by measuring the cGMP level in the hearts and plasma of BNP-treated infarcted mice [13]. 30 min after BNP injections, the cGMP level is increased in the hearts of treated mice and returns to normal 1 h after injection. In the circulation of treated mice, cGMP level is increased from 1 h to 24 h after BNP injections [13].

#### 4.2.5. Entresto Treatment

Myocardial infarction was induced in 8-week-old C57/BL6 mice (Figure 6A). Animals were randomly assigned into 3 different groups for treatment: H_2_O, LCZ696 (6 mg/kg/day), or LCZ696 (60 mg/kg/day). LCZ696 (6 mg/kg) corresponds to the dose used in the clinic in patients and LCZ696 (60 mg/kg) was already used on animal models and was shown to increase plasma ANP and BNP levels and angiogenesis [13,75]. In previous work, we demonstrated that the LCZ696 treatment (at both concentrations) does not affect body weight nor blood pressure and does not alter kidney functions [13]. We showed that only the high concentration of LCZ696 is associated with increased heart function and decreased heart remodeling 10 days after MI in mice. However, LCZ696 administration at both concentrations increases the cardiac vascularization in both areas of infarcted hearts [13].

LCZ696 drug was ground, dissolved in water and sonicated for 1 h before administration. The drug was administrated 24 h after MI and once daily for 10 days by oral gavage. BrdU (1 mg/mL) was added into the drinking water during the entire period of the experiment. Mice were sacrificed 10 days after MI.

### 4.3. Experimental Procedures at Sacrifice

At sacrifice, mice were weighted, the heart excised and weighted. The apex was embedded into OCT and slowly frozen for immunohistochemistry analysis. For quantitative RT-PCR (qRT-PCR) and/or Western blot analysis, the rest of the heart was quickly frozen in N_2_. Hearts from infarcted animals were separated into 3 zones for molecular and cellular analysis including CM isolation: (1) the infarction zone (ZI), (2) the border zone (BZ) and (3) the remote zone (RZ). The different areas were processed under microscopy: the ZI area is easily detectable by the color and the presence of the suture, the BZ corresponds to 0.5 mm tissue around the ZI and the rest of the heart is the RZ.

### 4.4. Adult CM Isolation and Counting

CM isolations were performed thanks to the Langendorff-Free method [38]. Rapidly after sacrifice, the chest of the mice was opened, descending aorta and inferior vena cava were cut, the aorta was clamped and EDTA buffer was perfused into the right ventricle. Thereafter, the heart was removed from the chest and the left ventricle was perfused with enzymatic solution containing collagenase II (120 U/mL), collagenase IV (120 U/mL) (Worthington, Biochemical Corporation, USA) and protease XIV (0.05 mg/mL or 0.175 U/mL) (Sigma-Aldrich, Merck, Darmstadt, Germany). In each experiment, a full digestion of the heart was obtained. After digestion, cardiac cells were dissociated by gentle pipetting, filtered and collected after 25 min sedimentation. For infarcted hearts, the tissue was separated into the three zones (ZI, BZ and RZ). The three different areas were weighted before gentle pipetting.

Total CM cell numbers were counted before sedimentation by two different people and in blinded experiments. The CMs were easily identified based on their size and on Troponin I expression (for all experiments with flow cytometry analysis). Round and rod-shaped CMs were counted and their numbers related to the heart, to the mg of tissue and also to the control hearts (saline-injected hearts) always performed in parallel the same day than BNP-treated hearts (i.e., with the same digestion buffer).

After 25 min sedimentation, CMs were used for molecular analysis, Western blot analysis, flow cytometry analysis or fixed with 2.5% PFA 15 min at room temperature (RT) and stained with DAPI to evaluate the frequency of mono and bi-nucleated CMs. Flow cytometry analysis based on Troponin I staining demonstrated that NMCs contamination in the CM fraction after sedimentation corresponds to ≤5% of the total cells and that sedimentation enriched the CM fraction into rod-shaped CMs (i.e., 90% of rod-shaped CMs were collected at the end of the sedimentation).

### 4.5. Determination of CM Size

CM area was assessed by α actinin and laminin stainings and image J software (1.53t version from the 24.08.2022, Mac OX X, Bethesda, MD, USA) for processing using CIF outliner cell Plugin. Only CMs with circularity >0.5 were considered.

### 4.6. Neonatal Cardiomyocyte Culture

Neonatal (1–2 days) C57BL/6 pups were sacrificed and CMs were isolated by enzymatic digestion according to the method previously described [9,12]. Briefly, the chest was opened and the heart removed. After separation of the atria from the rest of the heart, the ventricles were minced and digested with 0.45 mg/mL collagenase (Worthington, Biochemical Corporation, Lakewood, NJ, USA) and 1 mg/mL pancreatin (Sigma). After 3 rounds of digestion, cells were plated 2 times for 45 min in order to separate CMs (non-adherent cells) from the non-myocyte cells (NMCs). Thereafter, CMs were plated on gelatine (0.1%) coated plates and cultured in 3:1 mixture of DMEM and Medium 199 (Invitrogen Corp, San Diego, CA, USA) supplemented with 10% horse serum (Oxoid), 5% fetal bovine serum (Invitrogen), 10 mM Hepes, 100 U/mL penicillin G. To homogenize the experiment, 70,000 CMs were plated per well.

Immediately after plating, neonatal CMs were treated or not with BNP for 14 days. To inhibit NMC proliferation and to work on 95% pure CMs, cells were exposed the first 7 days of culture to cytosine-β-D-arabinofuranoside (AraC, 1 mg/mL). After 7 days of culture, AraC was removed until 14 days of culture. CMs were cultured in a hypoxia chamber (Stem Cell Technologies) flushed with 3% O_2_/5% CO_2_/92% N_2_ (Carbagas). Indeed, we previously observed that low oxygen concentration stimulates CM dedifferentiation and proliferation [9]. Medium was replaced 1–2 times/week.

CMs were treated with 3 different concentrations of BNP: 10 nM, 100 nM and 1000 nM and compared to untreated cells. Quantitative RT-qPCR, Western blot analysis as well as immunohistochemistry studies were performed after 14 days of culture.

### 4.7. Flow Cytometry Analysis

Adult CMs were isolated using enzymatic digestion (see above). Adult CMs isolated from Myh6 MerCreMer hearts were fixed with 5.5% formaldehyde and permeabilized with 0.5% saponin. CMs were stained during 20 min at RT with anti-troponin I antibody (Appendix A). Cells were analyzed with CytoFLEX (Beckmann Coulter) cytometer and data generated using FlowJo 10 software (10.7.1 version, Becton Dickinson and Compagny, Franklin Lakes, NJ, USA). The number of cTnI^+^ GFP^+^ cells per heart was obtained by relating the percentage of the cTnI^+^ GFP^+^ CMs acquired by flow cytometry analysis to the total number of CMs in the heart. Cells were stained with DAPI and only the DAPI negative cells (living cells) were analyzed.

### 4.8. Immunohistochemistry

Cells or OCT heart sections (5 µm-thick cryosections) were washed in PBS 1X and fixed in paraformaldehyde (2%) or formol (4%) for 10 min at RT. After 10 min of permeabilization (0.3% Triton x-100 in PBS) and one hour of blocking with normal serum, sections were probed with primary antibodies overnight at 4 °C (Appendix A). The second day, the secondary antibody was added on cells or heart sections. Slides were mounted with Dabco (Sigma) and pictures were captured with Nikon Eclipse TS100 or 90i microscope. Images were processed with Adobe Photoshop CC2015.

For the detection of BrdU incorporation, heart slides were fixed 10 min in 2% PFA, DNA was denatured 1 h at RT in HCl 2N before neutralization in Na Borate 0.1 M pH = 8.5 during 2 × 5 min. Then, sections were probed with primary antibodies overnight at 4 °C (Appendix A).

### 4.9. Western Blot Analysis

Total proteins were extracted from CMs isolated from adult infarcted and unmanipulated neonatal and adult hearts. Proteins were transferred to nitrocellulose membranes (Biorad) before incubation with primary antibodies overnight at 4 °C. Secondary antibodies were added 1 h at RT in the dark (Appendix A). The signal was detected and quantified with the Odyssey infrared imaging system (LI-COR. Biosciences, Bad Homburg, Germany). All results were normalized to the tubulin protein level or to the total protein level thanks to stainings with the Revert 700 Total protein Stain (LI-COR).

### 4.10. Time Lapse Microscopy

One day after birth, Myh6 MerCreMer/Tomato-EGFP pups were injected intraperitoneally with tamoxifen (1 mg/2 g). One day after, CMs were isolated and were plated on laminin (10 µg/mL) substrate. CMs were cultured in media containing 3:1 mixture of DMEM and Medium 199 (Invitrogen Corp, San Diego, CA, USA), supplemented with 10% horse serum (Oxoid), 5% fetal bovine serum (Invitrogen), 10 mM Hepes and, 100 U/mL penicillin G at 20% O_2_. After isolation, CMs were stimulated with BNP (100 nM). 48 h post plating, CMs were transferred to an OkoLab environment system with temperature and CO_2_ stage incubator. CMs expressing EGFP were targeted and images were taken each 40 min for 48 h using the Nikon Eclipse Ti2 inverted microscope and NIS-Elements software (4.51.01 version from 21.10.2017, Nikon Microscope Solutions, Tokyo, Japan).

### 4.11. Quantitative RT-PCR

qRT-PCR were performed using the SYBR Premix Ex Taq polymerase (Takara Bio Inc., Shiga, Japan) with the ViiA^TM^7 Instrument (Applied Biosystems, Waltham, MA, USA). Total RNA was isolated from heart tissue, CM cell cultures and isolated CMs using TRI-Reagent (Zymo Research, Orange County, CA, USA). cDNA was synthesized from RNA using PrimeScript RT reagent kit (Takara Bio Inc.). Polymerase chain reactions (PCR) were performed using the SYBR Premix Ex Taq polymerase (Takara Bio Inc.) with the ViiA^TM^7 Instrument (Applied Biosystems). Results were obtained after 40 cycles of a thermal step protocol consisting of an initial denaturation of 95 °C (1 s), followed by 60 °C (20 s) of elongation (α-skeletal actin has an elongation time of 30 s at 60 °C). The sequences of primers were reported in Appendix A. All results were normalized with the 18S housekeeping gene (Δ CT values). Means of ΔΔ CT (Δ CT_BNP–_Δ CT_saline_) values (versus untreated cells or NaCl treated mice) were calculated and results were represented as 2^−ΔΔCT^. Statistics were performed on ΔΔ CT individual values. SEM fold increase was calculated using 2^−ΔΔ CT high values^ − 2^−means of ΔΔCT^ [13].

### 4.12. Troponin Quantification in Plasma

Troponin T plasma levels were quantified using an electrochemiluminescence immunoassay analyzer “ECLIA” (cobas e 801 immunoanalysis system, Roche Diagnostics, Rotkreuz, Switzerland) in the hematology laboratory of the Centre Hospitalier Universitaire Vaudois (CHUV). Blood was collected 24 h and 3 days after infarction induction in heparin microvette (Sarstedt, Nümbrecht, Germany).

### 4.13. Statistical Analysis

All results were presented as means ± SEM. Statistical analyses were performed using the unpaired Student T test or Wilcoxon-Mann–Whitney tests. The alpha level of all tests was 0.05. The statistical analysis was performed only for the number of experiments ≥ 6.

## Figures and Tables

**Figure 1 cells-12-00007-f001:**
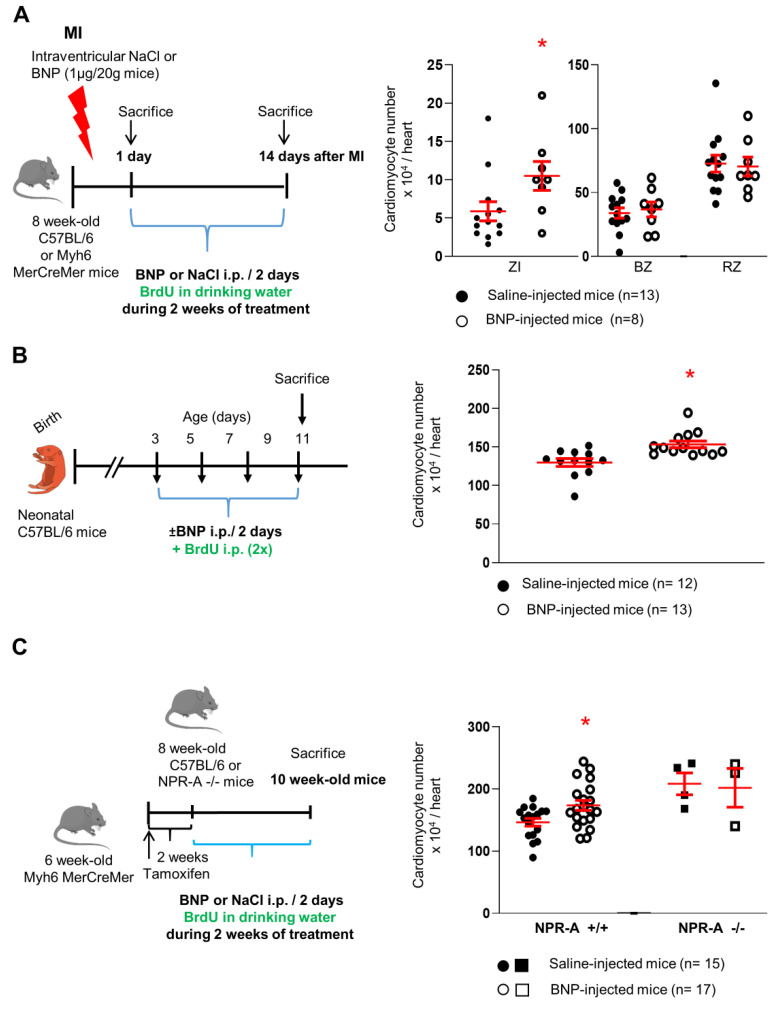
**BNP treatment leads to increased number of cardiomyocytes.** BNP injections in infarcted mice (**A**), neonatal (**B**) and adult (**C**) unmanipulated mice (experimental protocol as described in detail in Material and Methods section). For each experimental model, individual values (i.e., the total number of CMs in one heart) represented and the means ± SEM depicted in red. CMs isolated also from unmanipulated adult hearts deficient in NPR-A receptor (NPR-A −/−) and treated with saline (n = 4) or BNP (n = 3). ZI: infarct zone, BZ: border zone, RZ: remote zone. * *p* < 0.05.

**Figure 2 cells-12-00007-f002:**
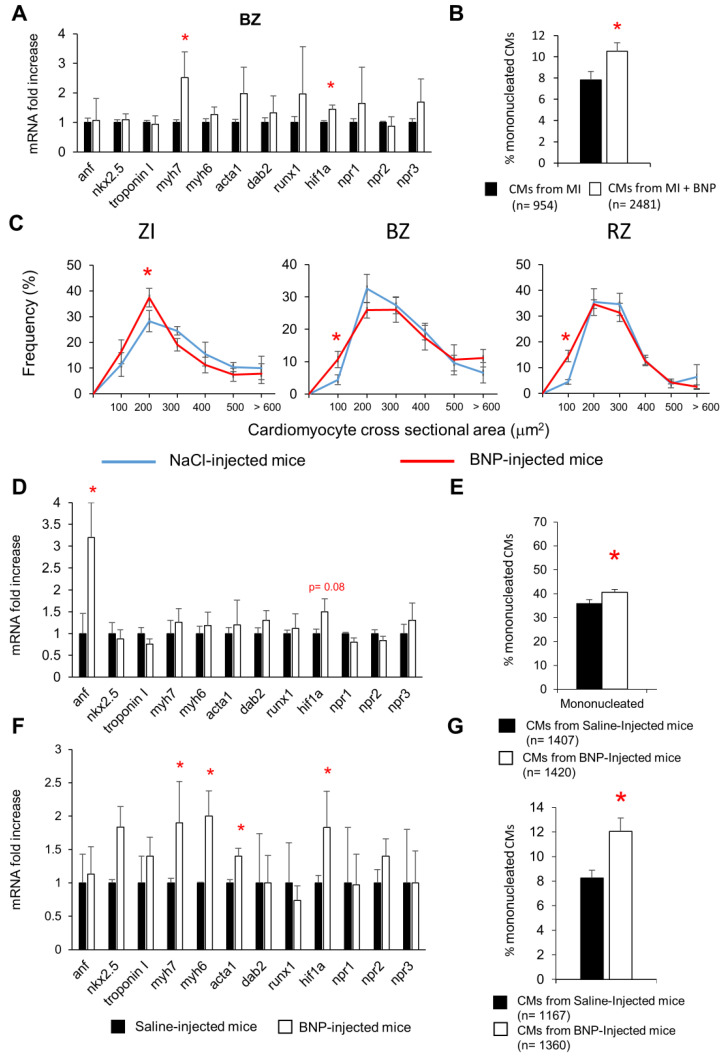
**Cardiomyocytes share common characteristics in BNP-treated hearts.** Cardiomyocytes were isolated from adult infarcted (**A**–**C**), unmanipulated neonatal (**D**,**E**) and adult (**F**,**G**) hearts treated with BNP (white) or saline (black). (**A**,**D**,**F**) Quantitative relative expression of mRNAs coding for different genes in CMs isolated from the BZ 10 days after surgery (**A**, n = 9 and 6 different CM isolations from saline- and BNP-treated infarcted hearts, respectively), from unmanipulated neonatal (**D**, n = 7 different CM isolations for both groups) and adult (**F**, n = 8 and 7 different CM isolations from saline- and BNP-treated hearts, respectively) hearts. Results expressed as fold-increase above the levels in CMs isolated from saline-injected infarcted mice. anf: atrial natriuretic peptide; myh7: β Myosin heavy Chain; myh6: α Myosin heavy Chain; acta1: skeletal muscle α actin; hif1α: hypoxic inducible factor 1 α; npr1, 2 or 3: natriuretic peptide receptor A, B or C, respectively. (**B**,**E**,**G**) Percentages of mononucleated CMs. CMs isolated from 6 different saline- and BNP-treated hearts. The total number of CMs evaluated in brackets. (**C**) Cardiomyocyte cross-sectional area frequency in the ZI, BZ and RZ of the infarcted hearts, obtained after immunostainings with antibodies anti-laminin and anti-actinin of 6 different saline- and BNP-treated hearts. At least 50 CMs measured in the ZI, 100 in the BZ and 300 in the RZ for each heart on different pictures. All results represented as means ± SEM. * *p* < 0.05.

**Figure 3 cells-12-00007-f003:**
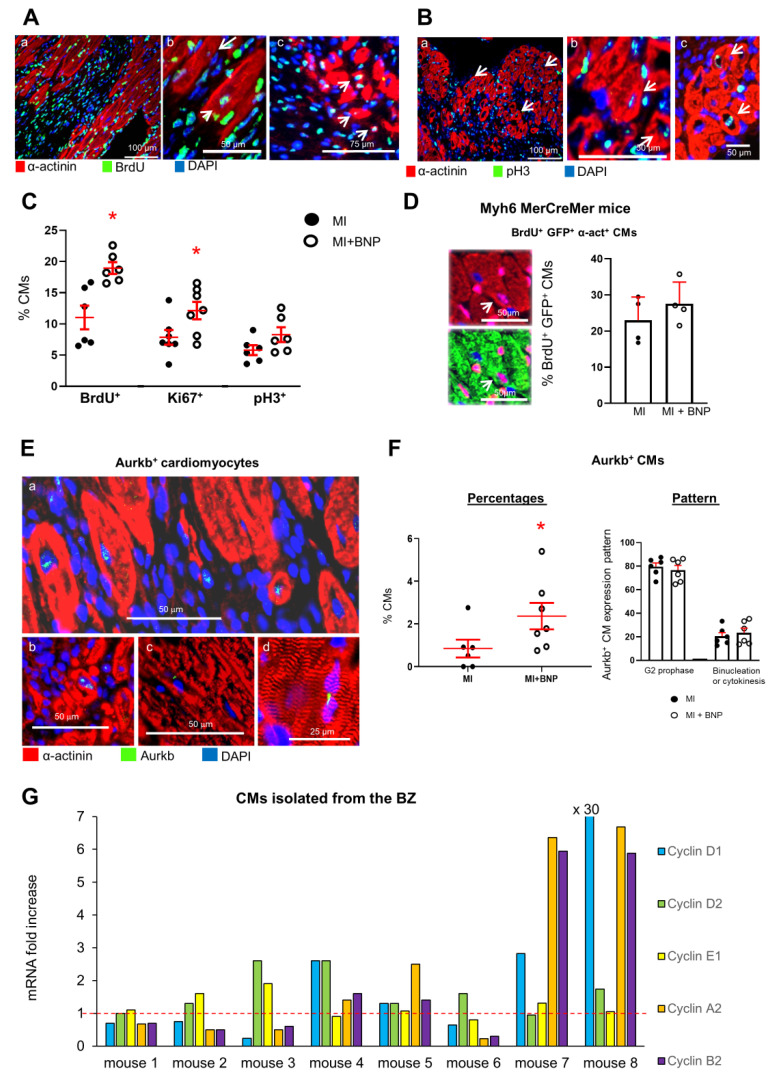
**BNP treatment stimulates cardiomyocyte cell cycle re-entry in the infarct and border area of the hearts 10 days after infarction.** (**A**,**B**) Representative pictures of immunostainings allowing the detection of CMs (i.e., α actinin^+^ cells) expressing BrdU or phospho histone 3 (pH3) in BNP-treated infarcted hearts. (**A**(**b**,**c**),**B**(**b**,**c**)): High magnification with examples of dedifferentiated CMs expressing BrdU or pH3. ×10: magnification. (**C**): Percentages of CMs (positive CMs/total number of α actinin^+^ cells) expressing proliferative markers in the ZI + BZ of infarcted hearts. For each heart, at least 10 pictures are analyzed. n = 6 different saline- and BNP-treated infarcted hearts. (**D**): BrdU incorporation assessed among GFP^+^ CMs (i.e., α actinin^+^ cells) in the infarction area of BNP-treated Myh6 MerCreMer hearts. Mice injected 2 weeks before infarct with tamoxifen in order to induce GFP expression specifically in CMs. At least 50 pictures were analyzed for saline- and BNP-treated mice originating from 4 different mice per group. The number of BrdU^+^ GFP^+^ CMs is related to the total number of CMs. (**E**): Representative pictures of immunostainings allowing the detection of CMs (i.e., α actinin^+^ cells) expressing the Aurora B proliferative marker in the ZI+BZ of BNP-treated hearts. Aurkb was mainly localized in the nuclei (**E**(**a**,**b**)) which characterized CMs undergoing prophase. CMs undergoing binucleation or cytokinesis were also detected (**E**(**c**,**d**)). (**F**) Percentages of CMs expressing Aurora B. For each heart, at least 10 pictures are analyzed. n = 6 different saline- and 7 BNP-treated infarcted hearts. Depending on Aurora B localization, the pattern of Aurkb^+^ CMs was determined on 6 different hearts per group. At least 40 Aurora B^+^ CMs are analyzed per heart. (**G**) mRNA expression coding for cyclin genes D1, D2, E1, A2 and B2. Results of CMs isolated from the BZ of 8 different BNP-treated infarcted hearts. Results expressed as the fold increase above the levels in CMs isolated from saline-treated infarcted hearts (represented by the red dotted line, n = 9 different saline-treated infarcted hearts). (**C**,**D**,**F**) Individual values represented and the means ± SEM represented in red. * *p* < 0.05.

**Figure 4 cells-12-00007-f004:**
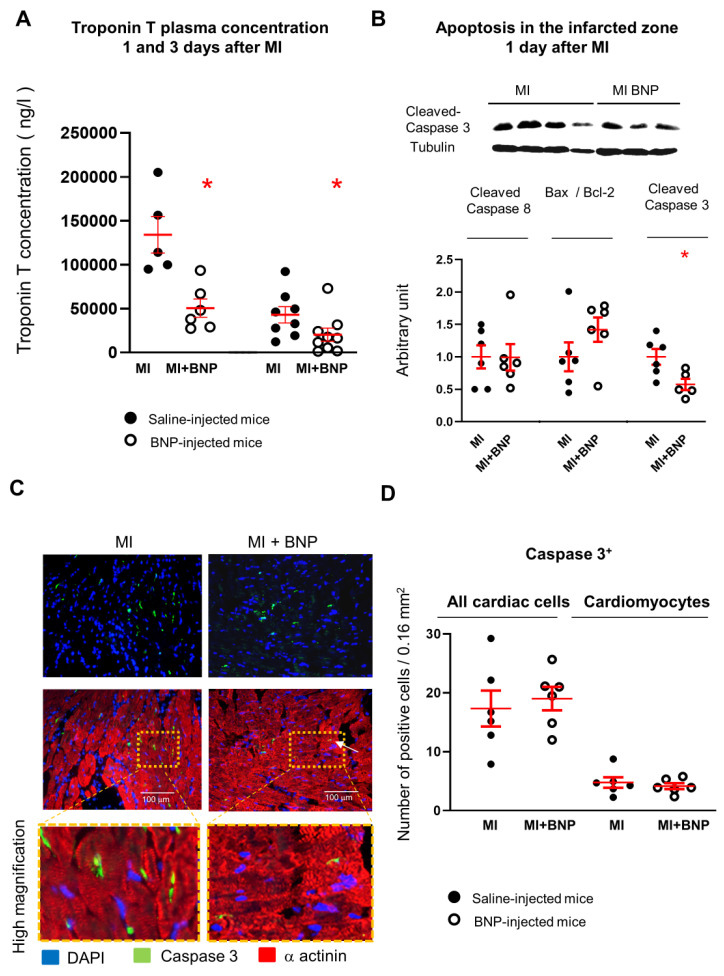
**BNP treatment protects cardiac cells in the ZI of infarcted hearts.** (**A**) Troponin T plasma concentration 1 and 3 days after myocardial infarction. Plasma were collected from 5 saline and 6 BNP-treated infarcted mice 1 day after MI and from 7 saline- and 8 BNP-treated infarcted mice 3 days after MI, respectively. (**B**) Representative Western blot of total proteins extracted from the infarcted area (ZI) of hearts treated with BNP (MI+BNP) or saline (MI) 1 day after surgery. Blots stained with antibodies against cleaved caspase-3 and Tubulin (used as loading control). Only the bands at the adequate molecular weight represented here: Tubulin (55 kDa), cleaved caspase-3 (12 kDa). Quantification of Western blot analysis for the cleaved caspase-8, the ratio Bax/Bcl-2 and cleaved caspase-3, all related to tubulin in BNP- and saline-treated hearts. n = 6 different saline- or BNP-infarcted hearts. Quantification of the data in BNP-treated infarcted hearts related to the average of saline-injected infarcted hearts. (**C**) Representative pictures of infarcted hearts injected or not with BNP and stained with antibodies against cleaved caspase-3 (green) and α-actinin (red). High magnifications depict apoptotic CMs in the ZI. (**D**) Graph represents the total number of cells and the number of cardiomyocytes expressing cleaved caspase-3 per picture in the infarction zone, one day after MI in BNP--- and saline-treated mice. Positive cells counted from at least 10 different pictures per heart and n = 6 saline- and 6 BNP-treated infarcted hearts. For all results, individual values represented and the means ± SEM depicted in red. * *p* ≤ 0.05.

**Figure 5 cells-12-00007-f005:**
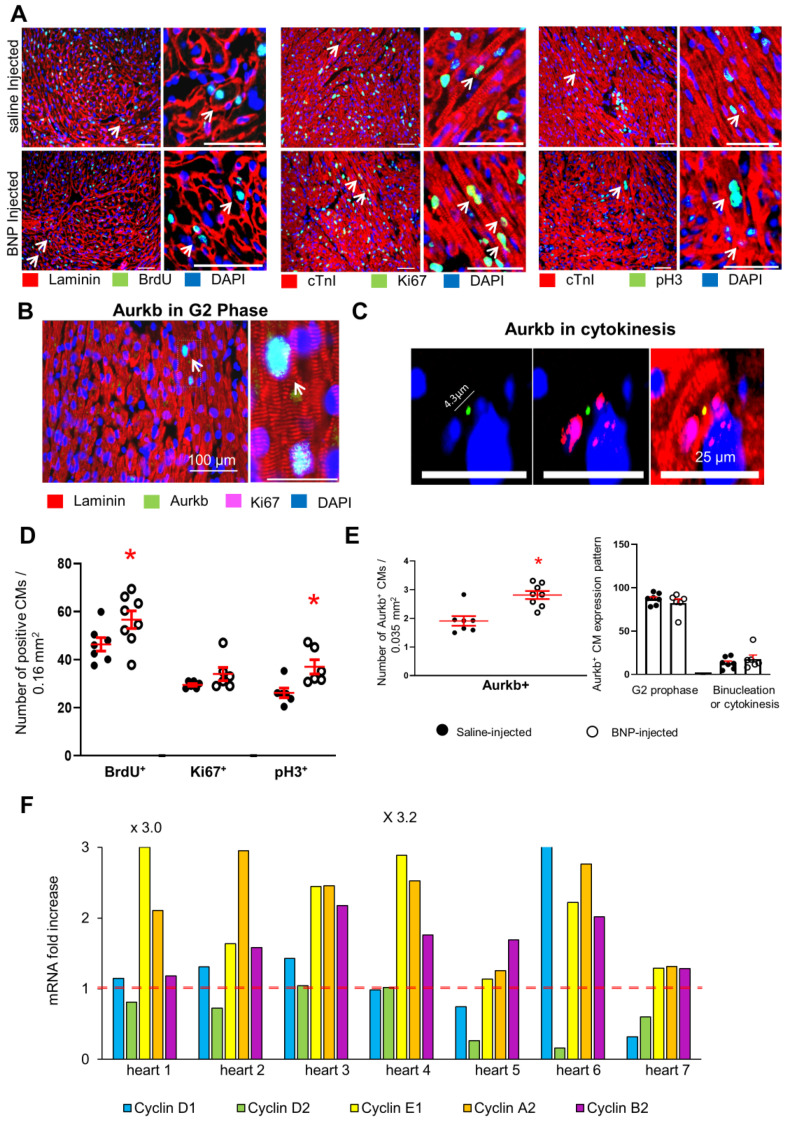
**BNP injections into neonatal mice improve cardiomyocyte proliferation.** (**A**–**C**) Representative pictures of immunostainings detecting CMs expressing BrdU, Ki67, phospho histone 3 (pH3) and Aurora B (Aurkb) in BNP and saline-treated neonatal hearts. Scale bars = 100 µm in (**A**,**B**) and 25 µmm in (**C**). (**D**,**E**) Number of CMs per picture expressing these proliferative markers. At least 10 different pictures are analyzed per heart. n = 7 and 8 different saline-or BNP-injected hearts, respectively. **(E)** Within CMs expressing the Aurora B protein, the percentage of CMs in cytokinesis, binucleation or prophase was established. At least 70 Aurkb^+^ CMs are analyzed per heart. Results are the means of the results obtained for 7 and 6 saline or BNP-injected hearts, respectively. (**F**) Expression of mRNAs coding for cyclin D1, D2, E1, A2 and B2. Results of CMs isolated from neonatal BNP-treated hearts expressed as a fold increase above the levels in CMs isolated from saline-treated hearts (represented by the red dotted line, n = 7 different isolations from saline- injected neonatal hearts). Seven different CM cell isolations from BNP-treated hearts represented. (**D**,**E**) individual values represented and the means ± SEM depicted in red. * *p* ≤ 0.05.

**Figure 6 cells-12-00007-f006:**
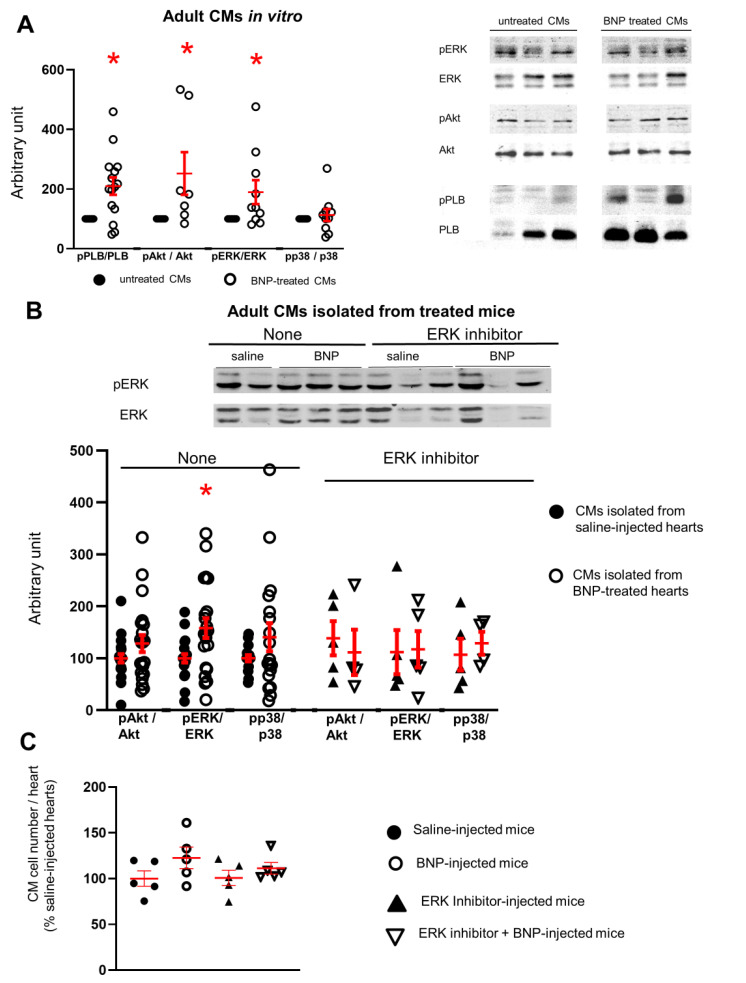
**The ERK MAP kinase signaling pathway is activated in cardiomyocytes after BNP treatment in vitro and in vivo.** (**A**,**B**) Measurement of phospho phospholamban (pPLB)/phospholamban (PLB), pAkt/Akt, pERK/ERK and pp38/p38 ratio by Western blot analysis in adult CMs stimulated or not with BNP for 1-6h in vitro (**A**), (n = 7–14 different CM isolations) or in CMs isolated from unmanipulated adult hearts injected with saline or BNP for 2 weeks (**B**) **left**, (n = 20 saline and 21 BNP-injected hearts). (**B**), **right**: CMs were also isolated from adult unmanipulated hearts treated with ERK inhibitor (PD0325901) 1h before saline or BNP injections. n = 5 different mice per group. Total protein stainings used as loading control (Appendix A) and quantification of all the results obtained related to the average of untreated CMs (black circle for **A**) or of CMs isolated from saline-injected hearts (**B**), (black circles). (**C**) Number of CMs isolated from adult unmanipulated hearts treated or not with ERK inhibitor before BNP or saline injections. Mice (n = 5 per group) sacrificed after 2 weeks of BNP treatment. The number of CMs in BNP-treated hearts related to the numbers found in saline-injected hearts. Individual values represented and the means ± SEM depicted in red. * *p* < 0.05 versus untreated CMs or CMs isolated from saline-treated hearts.

**Figure 7 cells-12-00007-f007:**
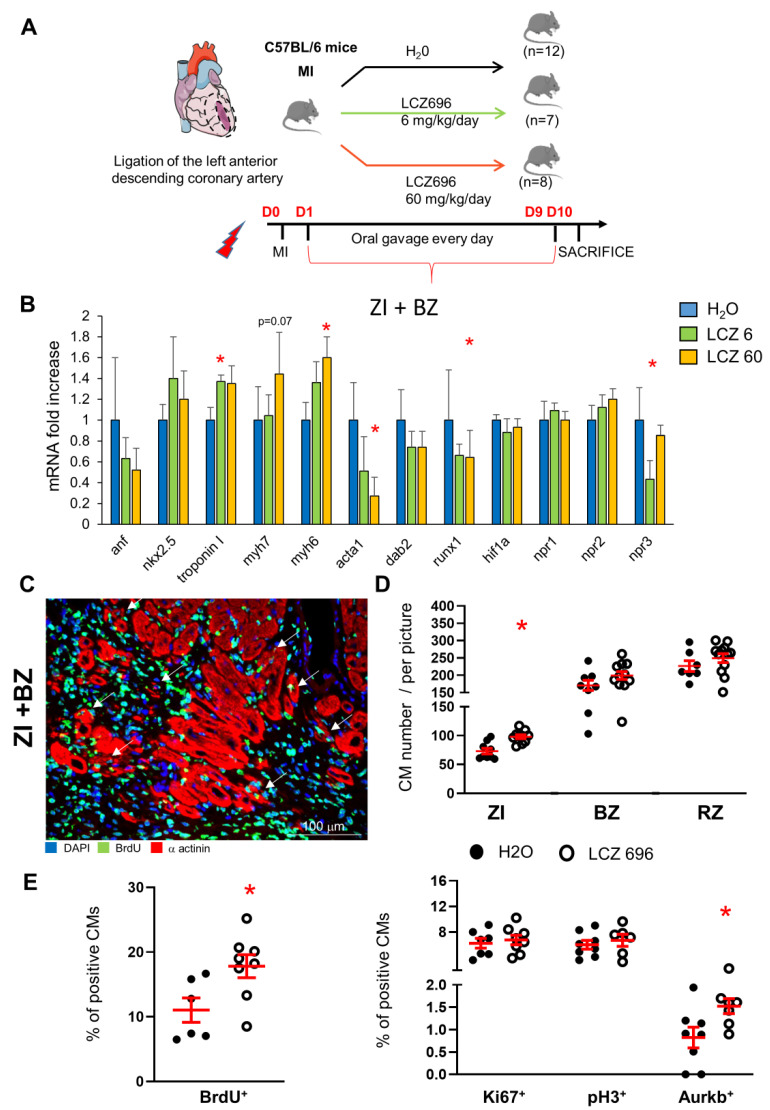
**LCZ696 treatment increases the cardiomyocyte number in the infarct and border area of infarcted hearts.** (**A**) Experimental protocol as described in detail in the Material and Methods section. (**B**): Quantitative relative expression of mRNAs coding for cardiomyocyte specific genes in the border zone of the infarcted hearts treated or not with LCZ696 at two different doses for 10 days. ZI: infarct zone, BZ: border zone, RZ: remote zone. Results expressed as fold-increase above the levels in untreated infarcted hearts (H_2_O). Results represented as means ± SEM; n = 12 for H_2_O treated mice, n = 7 for LCZ696 6 mg/kg/day and n = 8 for LCZ696 60 mg/kg/day treated mice. anf: atrial natriuretic peptide; myh7: β Myosin heavy Chain; myh6: α Myosin heavy Chain; acta1: skeletal muscle alpha-actin; hif1α: hypoxic inducible factor 1α; npr1, 2 or 3: natriuretic peptide receptor A, B or C, respectively. (**C**) Representative picture of immunostainings using antibodies against α actinin and BrdU of the infarct and border area of a LCZ696-treated hearts. White arrows depict BrdU positive cardiomyocytes, which are dedifferentiated small CMs. (**D**) CM number counted in the different area of the infarcted hearts treated or not with LCZ696 (no difference between both doses, the results are pooled). At least 10 different pictures counted per area and heart. n = 7–9 H_2_O- and 9–12 LCZ696-treated infarcted hearts. Number of CMs in LCZ696-treated hearts related to the numbers found in H_2_O-treated infarcted hearts. (**E**) Percentages of CMs expressing BrdU, Ki67, pH3 and Aurora B in the infarcted and border area of the injured hearts. At least 10 different pictures counted per mouse. n = 6–8 H_2_O-or LCZ696-treated infarcted hearts. (**D**,**E**) Individual values represented and the means ± SEM depicted in red. * *p* < 0.05 versus untreated infarcted hearts.

## Data Availability

The original data will be provided on request.

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
