# Peer review of "Brain Natriuretic Peptide Protects Cardiomyocytes from Apoptosis and Stimulates Their Cell Cycle Re-Entry in Mouse Infarcted Hearts"

_cells, 2022, doi:10.3390/cells12010007_

Round 1
Reviewer 1 Report
Present study by Bon-Mathier A et al investigated that BNP contributes to adult heart homeostasis in physiological conditions and to the protection of the adult hearts after MI. They showed that BNP treatment leads to increased CM cell number by protecting the CMs and revealed detailed mechanisms of action. The study is well-designed using wide –range of methodology. There are some questions to clarify some details.
Comments:
1. The abstract of present MS miss the highly important details regarding to animal models and experimental methodology. Please complete.
2. The chosen dosage of BNP and neprilisin inhibitor should be explained with previous references.
3. Why authors design BNP treatment with intracavital injection first and then shift to intraperitoneal.
4. Kinetic data of BNP should be added to MS.
5. Individual values should be presented in each bar graph.
6. Western blot evaluation should be presented in blot form for each investigated targets.
7. BNP action and its second messenger signalling pathways, like phopholamban phosphorylation should be discussed with references. Please add PMID: 25022512, PMID: 20349314
8. Ratio of oxygen in hypoxic chamber should added to methods section.
9. Adult cardiac myocytes cultured for long period undergo severe de-differentiation. Did this issue affected present investigations?
10. Did authors could show infarct size data strengthening the reproducibility and reliability of LAD occlusion?
11. Mortality data are missing, please add.
Reviewer 2 Report
The manuscript by Bon-Mathier interrogates the role of BNP in their previously reported findings that it is beneficial in animal models of MI.
The first paragraph of the results (highlighting results from the authors' previous study) should be moved to the introduction - this is important background to this present work rather than results
Figures with mRNA expression - Figure 2 shows a mean sd but the following figures then show per individual heart - this demonstrates the variability of expression between individual animals. Perhaps add the individual values for each mouse within the barchart to figure 2D and 2F to highlight this
Figure 3 (apoptosis) it would be useful to show a further marker of apoptosis in addition to cleaved caspase 3
Figure 4 and Figure 3 are mis-labelled
